# Analogue closed-loop optogenetic modulation of hippocampal pyramidal cells dissociates gamma frequency and amplitude

**Elizabeth Nicholson[1†], Dmitry A Kuzmin[1†], Marco Leite[1], Thomas E Akam[2‡], Dimitri Michael Kullmann[1]***

[1]UCL Institute of Neurology, University College London, London, United Kingdom; [2]Champalimaud Neuroscience Program, Champalimaud Center for the Unknown, Lisbon, Portugal

**\*For correspondence:**
d.kullmann@ucl.ac.uk

[†]These authors contributed equally to this work

**Present address:** [‡]Department of Experimental Psychology, University of Oxford, England, United Kingdom

**Competing interests:** The authors declare that no competing interests exist.

**Abstract** Gamma-band oscillations are implicated in modulation of attention, integration of sensory information and flexible communication among anatomically connected brain areas. How networks become entrained is incompletely understood. Specifically, it is unclear how the spectral and temporal characteristics of network oscillations can be altered on rapid timescales needed for efficient communication. We use closed-loop optogenetic modulation of principal cell excitability in mouse hippocampal slices to interrogate the dynamical properties of hippocampal oscillations. Gamma frequency and amplitude can be modulated bi-directionally, and dissociated, by phase-advancing or delaying optogenetic feedback to pyramidal cells. Closed-loop modulation alters the synchrony rather than average frequency of action potentials, in principle avoiding disruption of population rate-coding of information. Modulation of phasic excitatory currents in principal neurons is sufficient to manipulate oscillations, suggesting that feed-forward excitation of pyramidal cells has an important role in determining oscillatory dynamics and the ability of networks to couple with one another.
DOI: https://doi.org/10.7554/eLife.38346.001

## Introduction

Gamma-band (approximately 30 to 120 Hz) oscillations have been implicated in the modulation of attention and perception, in action initiation, spatial navigation and memory encoding, and have also been proposed to underlie flexible information routing among anatomically connected regions (*Akam and Kullmann, 2010*; *Akam and Kullmann, 2014*; *Börgers and Kopell, 2003*; *Fries, 2005*; *Kirst et al., 2016*; *Lisman, 2010*; *Rodriguez et al., 1999*; *Salinas and Sejnowski, 2001*; *Schnitzler and Gross, 2005*). Central to several of these proposed roles is the ability of gamma oscillations in different areas to enter into, and exit, states of synchrony with one another (*Akam et al., 2012*; *Fries, 2015*; *Varela et al., 2001*). Evidence for behavioral-state-dependent coupling and uncoupling comes from variable oscillatory coherence among distinct components of the visual cortex, correlating with selective stimulus attention (*Bosman et al., 2012*; *Grothe et al., 2012*). An earlier study in the rodent hippocampal formation showed that the CA1 subfield can flip between a state of coherence with the medial entorhinal cortex at ~110 Hz and a state of coherence with the CA3 subfield at ~40 Hz, correlating with information flow through the temporo-ammonic and Schaffer collateral pathways, respectively (*Colgin et al., 2009*). Although several experimental confounds cloud the interpretation of coherence measured from local field potential (LFP) recordings

(*Buzsáki and Schomburg, 2015*), these studies provide some of the most compelling evidence that gamma-band oscillatory entrainment underlies flexible functional connectivity.

Although the cellular mechanisms underlying gamma oscillations have been extensively studied (*Bartos et al., 2007*; *Buzsáki and Wang, 2012*), there remain uncertainties over the fundamental determinants of their dynamics and the relative contributions of excitatory and inhibitory signalling. Gamma-band oscillations can be induced in vitro in the presence of blockers of ionotropic glutamate receptors (*Whittington et al., 1995*), or in vivo by optogenetic stimulation of parvalbumin-positive interneurons (*Cardin et al., 2009*; *Sohal et al., 2009*), underlining the importance of fast perisomatic inhibition (*Bartos et al., 2002*; *Fisahn et al., 2004*; *Mann et al., 2005*). Robust population oscillations can also be simulated in exclusively inhibitory networks (*Wang and Buzsáki, 1996*). These experimental and computational observations emphasize the importance of inhibitory kinetics. Nevertheless, gamma-band oscillations can be entrained by sinusoidal optogenetic stimulation of pyramidal neurons in an in vitro hippocampal slice preparation (*Akam et al., 2012*). This observation implies that phasic depolarization of principal cells can determine the gamma rhythm and argues against a model where the only role of pyramidal cells is to tonically depolarize a network of reciprocally coupled interneurons (*Bartos et al., 2007*; *Tiesinga and Sejnowski, 2009*).

Further insight into the dynamical mechanisms of synchronization between oscillating networks comes from examining the phase response curve (PRC) of the network oscillation, defined as the phase advance or delay produced by a transient stimulation, as a function of the instantaneous phase at which the stimulus is delivered. The finding that gamma in an in vitro hippocampal slice preparation shows a biphasic PRC (*Akam et al., 2012*) is consistent with the hypothesis that this oscillation can be entrained by appropriately modulated afferent activity. The shape of the PRC is furthermore accurately reproduced with a simple neural mass model (*Wilson and Cowan, 1972*), where extracellular electrical or optogenetic stimuli are represented as transient perturbations of the instantaneous level of excitation or inhibition (*Akam et al., 2012*). Recent theoretical work has derived population PRCs for oscillations in spiking network models, providing an insight into how mechanisms of oscillation generation determine entrainment properties (*Akao et al., 2018*; *Kotani et al., 2014*). Nevertheless, there remains a large gap between the PRC and understanding the determinants of the oscillatory frequency and interactions between gamma-generating circuits.

The present study investigates the dynamical properties of gamma oscillations by using closed-loop optogenetics to create an artificial feedback loop between the oscillatory network activity (as assessed by the LFP) and excitatory input to the principal cell population. Specifically, we delivered analogue-modulated excitation whose strength was a function of the instantaneous phase and amplitude of the oscillation. This approach is quite distinct from previous closed-loop applications of optogenetics (*Grosenick et al., 2015*), which have adopted one of four main strategies. First, several studies have used the detection of a change in the state of a network, such as the onset of an electrographic seizure (*Krook-Magnuson et al., 2013*; *Paz et al., 2013*) or sharp-wave ripple (*Stark et al., 2014*), to trigger light delivery and return the network to its ground state. Second, light pulses have been timed according to the phase of a theta oscillation (*Siegle and Wilson, 2014*), while examining the consequences for behaviour. In the latter example, the theta oscillation itself was not altered. Third, optogenetics has been used to regulate the overall activity of a population of neurons at a desired level (*Newman et al., 2015*). Fourth, optogenetic depolarization of interneurons, triggered by spikes in an individual principal cell, has been used to simulate a feedback inhibitory loop to interrogate their role in gamma (*Sohal et al., 2009*; *Veit et al., 2017*). The goal of the present investigation is qualitatively different: to understand how the spectral characteristics of gamma are affected by rhythmic excitation arriving at different phases. Computational simulations have suggested that closed-loop optogenetics could be used to adjust the phase of gamma (*Witt et al., 2013*), but whether it can alter its frequency or amplitude remains unclear.

## Results

### Closed-loop feedback modulation of affects gamma oscillations in CA1

We expressed the red-shifted optogenetic actuator C1V1 (*Yizhar et al., 2011*) in the mouse hippocampus CA1 under the *Camk2a* promoter to bias expression to excitatory neurons. The local field potential (LFP) was recorded in the CA1 pyramidal cell layer in acute hippocampal slices. A slowly

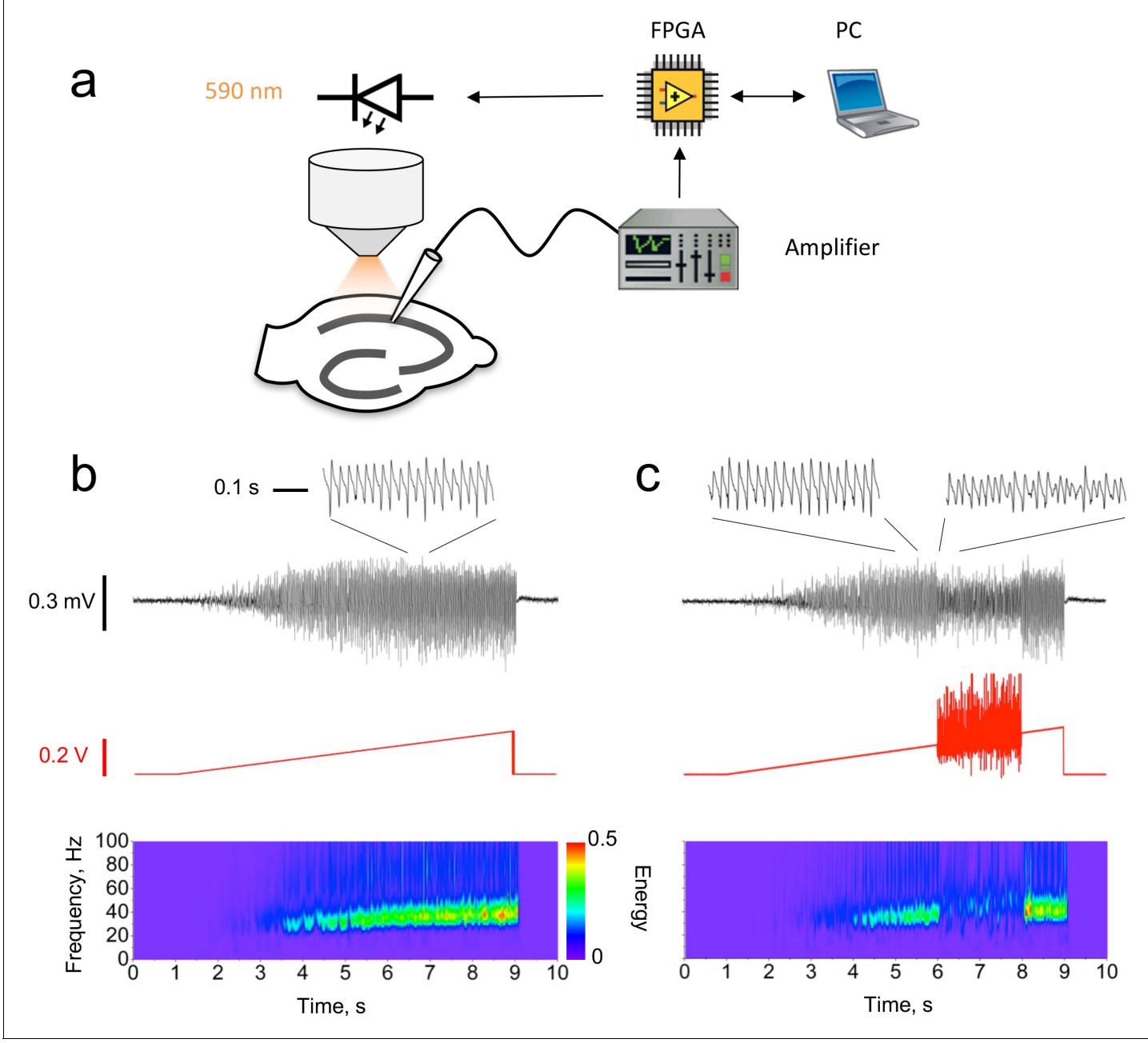

**Figure 1.** Closed-loop modulation of gamma oscillation. (a) Experimental design. The LFP in CA1 was used to modulate a ramp command generated by the PC via a field-programmable gate array (FPGA). The modulated ramp voltage command was then passed to the light-emitting diode (LED) driver, which implemented a threshold-linear voltage-to-current conversion. (b) Unmodulated oscillation recorded in CA1 induced by a linear ramp LED driver command. Black trace: LFP with an expanded section showing the characteristic shape of the gamma oscillation (inset). Red trace: LED ramp command. Bottom: LFP Morlet wavelet spectrogram. (c) Closed-loop oscillation clamp applied between 6 and 8 s, obtained by multiplying the ramp command by $(1 + k_1 LFP + k_2 dLFP/dt)$, with $dLFP/dt$ averaged over 2 ms intervals. For this example, $k_1 = 0$ mV$^{-1}$, $k_2 = 25$ ms mV$^{-1}$. The oscillation amplitude was reduced by approximately 60% (insets), with no net change in frequency.

DOI: https://doi.org/10.7554/eLife.38346.002

The following source data is available for figure 1:

**Source data 1.** *Figure 1* source data
DOI: https://doi.org/10.7554/eLife.38346.003

increasing ramp of light (peak wavelength 590 nm) was delivered via a light-emitting diode (LED) coupled to the epifluorescence port of an upright microscope, eliciting a gamma oscillation

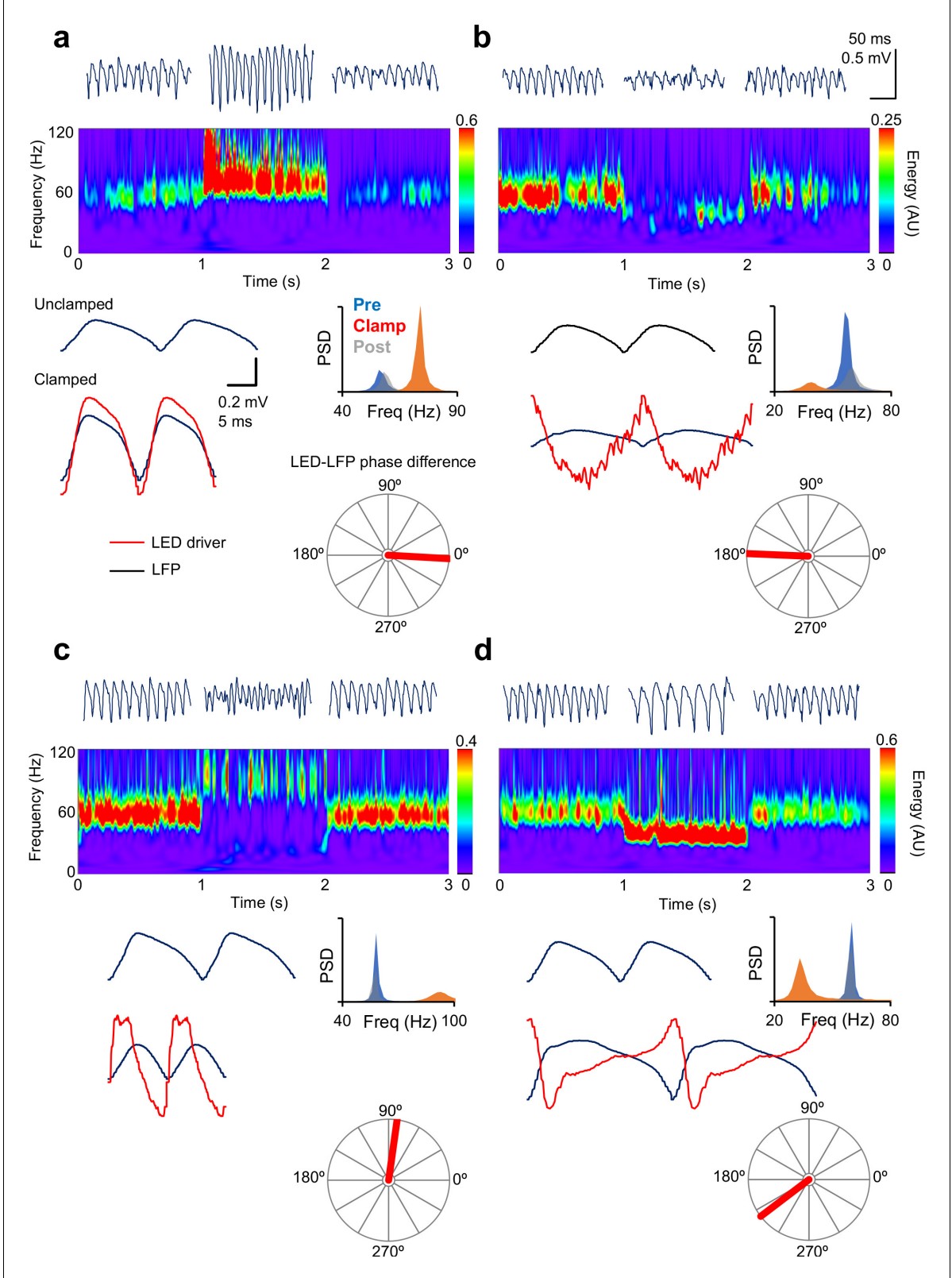

**Figure 2.** Gamma clamp allows bidirectional modulation of frequency and power. (a) In-phase modulation led to an increase in gamma power and frequency. Top: 200 ms-long segments of the LFP before, during and after closed-loop modulation of the LED driver. Middle: spectrogram. Bottom, left: two cycles of the average oscillation before and during the oscillation clamp. The average LED command (red trace, arbitrary scale) is shown superimposed on the clamped oscillation. Right: power spectral density before (blue), during (red) and after (grey) clamp. The polar plot shows the
*Figure 2 continued on next page*

*Figure 2 continued*

phase relationship between the LED command and the LFP. (**b**) Anti-phase modulation led to decreases in both frequency and power. (**c**) An increase in oscillation frequency, together with a decrease in power, was obtained with ~90° phase-advance of the LED driver command over the LFP. (**d**) A decrease in frequency, together with an increase in power, was obtained when the LED modulation was delayed relative to the LFP by ~145°. Scale bars apply to all panels.

DOI: https://doi.org/10.7554/eLife.38346.004

The following source data is available for figure 2:

**Source data 1.** *Figure 2* source data

DOI: https://doi.org/10.7554/eLife.38346.005

(*Figure 1a,b*), as previously reported in rodents (*Adesnik, 2018*; *Adesnik and Scanziani, 2010*; *Akam et al., 2012*; *Butler et al., 2016*; *Pastoll et al., 2013*), cats (*Ni et al., 2016*) and monkeys (*Lu et al., 2015*).

In order to investigate the role of phasic excitation in setting the dynamical properties of gamma, we used the LFP itself to manipulate the optogenetic drive in real time. The LED driver command was multiplied by a simple function of the instantaneous value of the LFP and its time-derivative: $(1 + k_1\,\text{LFP} + k_2\,d\text{LFP}/dt)$, where $k_1$ and $k_2$ are positive or negative constants. These operations were implemented with a field-programmable gate array (FPGA) and applied for a defined duration (typically 1 or 2 s) during the ramp. This yielded a change in the spectral properties of the oscillation, which lasted for the duration of the closed-loop feedback (*Figure 1c*). Because both the LFP and its time-derivative fluctuated about 0, the 'gamma clamp' had little effect on the average illumination intensity relative to an unmodulated ramp. Changes in the oscillation frequency or power could therefore not be attributed to a net increase or decrease in the average optogenetic drive to pyramidal neurons.

We adjusted the clamp function by altering the values of $k_1$ and $k_2$ and asked whether the frequency and/or power of the gamma oscillation can be modulated bidirectionally. Changes in spectral properties were related to the phase difference between the LFP and the LED drive during the clamp, as estimated from the cross-spectrum at maximal magnitude. In-phase modulation, achieved by setting $k_1$ positive and $k_2 = 0$, led to an increase in oscillatory power and frequency (*Figure 2a*). Modulating the ramp in anti-phase relative to the LFP, by setting $k_1$ negative, led to a decrease in both frequency and power (*Figure 2b*). Advancing the phase of the clamp by approximately 90°, achieved by setting $k_1 = 0$ and $k_2$ positive, increased the frequency of the oscillation whilst decreasing its power (*Figure 2c*). Finally, a decrease in frequency and increase in power was achieved by delaying the trough of the clamp modulation relative to the LFP, by setting $k_2$ negative (*Figure 2D*). Detailed inspection of the ramp command waveform during the clamp shows that it was in some cases distorted relative to the LFP, as expected from its non-sinusoidal shape (*Cole and Voytek, 2017*) (e.g. *Figure 2c,d*), and so the LED-LFP phase differences were only approximate.

Attempts to estimate the instantaneous oscillation phase, for instance using a Hilbert transform, and to use this to phase-advance or phase-delay a template of the LFP, compressed or stretched in time, were unsuccessful: the phase jitter and cycle-to-cycle variability in the amplitude and frequency of the gamma oscillation (see LFP traces in *Figure 2*) prevented accurate estimation of these parameters in the face of closed-loop feedback.

## Oscillation clamp is broadly consistent with the phase response curve of gamma

Changes in frequency and power, expressed in relation to the approximate phase difference between the LED command and the LFP, were qualitatively consistent across experiments (*Figure 3a–c*). Moreover, as the LED-LFP phase difference was rotated through a complete cycle, the effect on the oscillation in the two-dimensional plane defined by the change in oscillation frequency and power also rotated through 360°, such that with the appropriate phase of closed-loop feedback the network oscillation could be pushed in any desired direction in the oscillation frequency-power space (*Figure 3c*).

To gain a mechanistic insight, we asked if the characteristic relationship between the frequency change and the LED-LFP phase difference could be explained by the shape of the phase-response

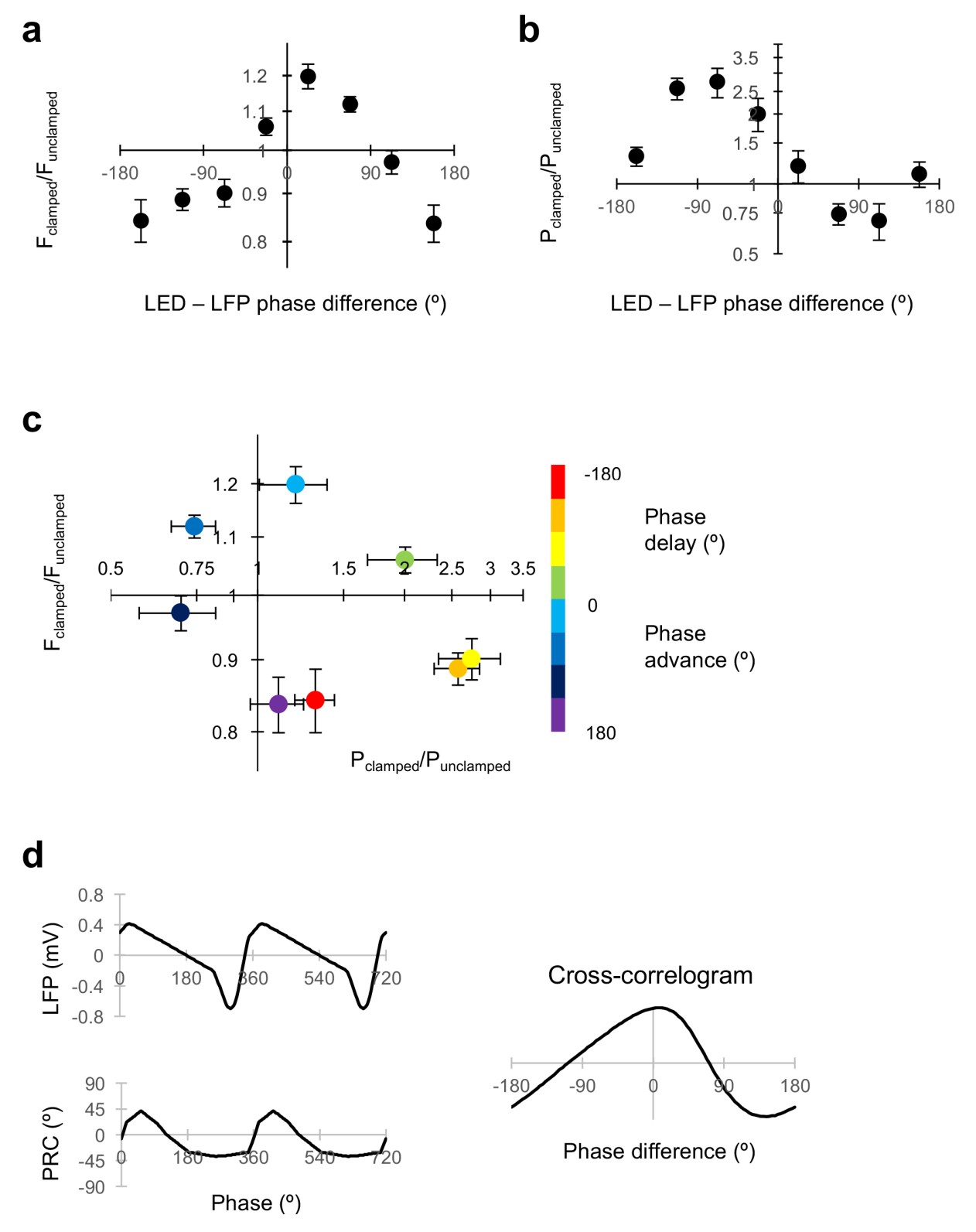

**Figure 3.** Dissociable modulation of oscillation frequency and power. (a) Dependence of frequency change on the phase relationship between the LED modulation and the LFP (positive values indicate LED phase advance relative to LFP). Changes in frequency are plotted as $F_{clamped}/F_{unclamped}$, where the unclamped frequency was averaged from the gamma oscillation for 1 s before and 1 s after the gamma clamp was applied. Data are shown as mean ±SEM (n = 19 experiments). A positive phase difference indicates that the modulation was phase-advanced relative to the LFP. (b) Dependence

*Figure 3 continued on next page*

*Figure 3 continued*

of power change on the phase difference plotted as in (a). (**c**) Change in frequency plotted against change in power for different LED – LFP phase differences (colour code at right). (**d**) Average LFP and phase response (PRC) curve from *Akam et al. (2012)* (left). The circular cross-correlogram at right yields a prediction of the effect of a continuous modulation on the oscillation phase, and therefore on its frequency, in rough agreement with the observed relationship in (a).

DOI: https://doi.org/10.7554/eLife.38346.006

The following source data is available for figure 3:

**Source data 1.** *Figure 3* source data
DOI: https://doi.org/10.7554/eLife.38346.007

curve (PRC) previously reported (*Akam et al., 2012*). In that study, a brief 'kick' was applied on top of the LED ramp command, and the phase advance or delay of subsequent oscillations was related to the phase of the LFP at which the transient occurred. A phase delay was observed when the transient optogenetic stimulus was delivered at the trough of the LFP, when pyramidal neurons are most likely to fire. The maximal phase advance, in contrast, occurred when the stimulus was delivered approximately one third of a cycle after the trough of the LFP. Assuming linear behaviour, the effect of modulating the light intensity in closed loop can be obtained by averaging the product of the phase shift and the LFP over the entire cycle of the oscillation. The circular cross-correlogram between the typical LFP shape and the PRC should then predict the effect of modulating the optogenetic drive by the shape of the LFP itself at arbitrary degrees of phase advance or delay (*Figure 3d*). In-phase modulation is expected, on the basis of this calculation, to phase-advance the oscillation, and thus to result in an increase in oscillatory frequency over successive cycles. Anti-phase modulation, in contrast, is predicted to phase-delay the oscillation, and thus to decrease its frequency. The circular cross-correlation is, moreover, asymmetrical, broadly consistent with the shape of the relationship between the change in frequency and LED-LFP phase difference observed in the clamp experiments (*Figure 3a*).

Although the shape of the PRC is consistent with the changes in gamma frequency achieved with closed-loop modulation at different LED-LFP phase differences, on its own it says nothing about changes in power. Power was maximally decreased with a phase advance of the LED command over the LFP around 90°, whilst it was maximally increased with a phase delay around 90° (*Figure 3b*). The relative phases at which frequency and power were altered are, however, consistent with the behaviour of a normal form description of a super-critical Hopf bifurcation in the vicinity of its limit-cycle. In this scenario, the LFP would approximate an observed variable, and the optogenetic drive would act in the direction of a hidden variable at a +90° angle to the LFP.

A deeper understanding of the characteristic changes in frequency and power of the LFP with different LED-LFP phase differences requires an insight into how neurons spike and ultimately how membrane currents respond to the fluctuations in optogenetic drive. We therefore performed single-cell recordings in parallel with the LFP recordings.

## Gamma clamp affects the timing, not rate, of pyramidal neuron firing

Although the average illumination intensity was not altered during the gamma clamp, for certain LED-LFP phase relationships gamma power increased or decreased robustly. Inhibitory currents in principal neurons, rather than spikes or excitatory currents, have previously been shown to be the main determinant of the LFP (*Oren et al., 2010*), suggesting that the change in power during the clamp is not a direct effect of the optogenetic drive but results instead from a change in pyramidal neuron synchrony or phase, in a reciprocal relationship with the degree and temporal synchrony of interneuron recruitment. To determine how the clamp affects pyramidal neuron firing, we repeated experiments with an additional patch pipette to record from individual pyramidal neurons in cell-attached mode. Individual action potentials were used to align the simultaneously recorded LFP, and to estimate the phase at which they occurred. During an unmodulated ramp, pyramidal cells tended to spike sparsely, close to the trough of the oscillation, consistent with previous studies of pharmacologically induced oscillations (*Fisahn et al., 1998*). During the clamp, an increase in oscillatory power was associated with a corresponding increase in the degree of synchrony of pyramidal cell firing: the circular dispersion of LFP phase at which pyramidal cells fired decreased relative to

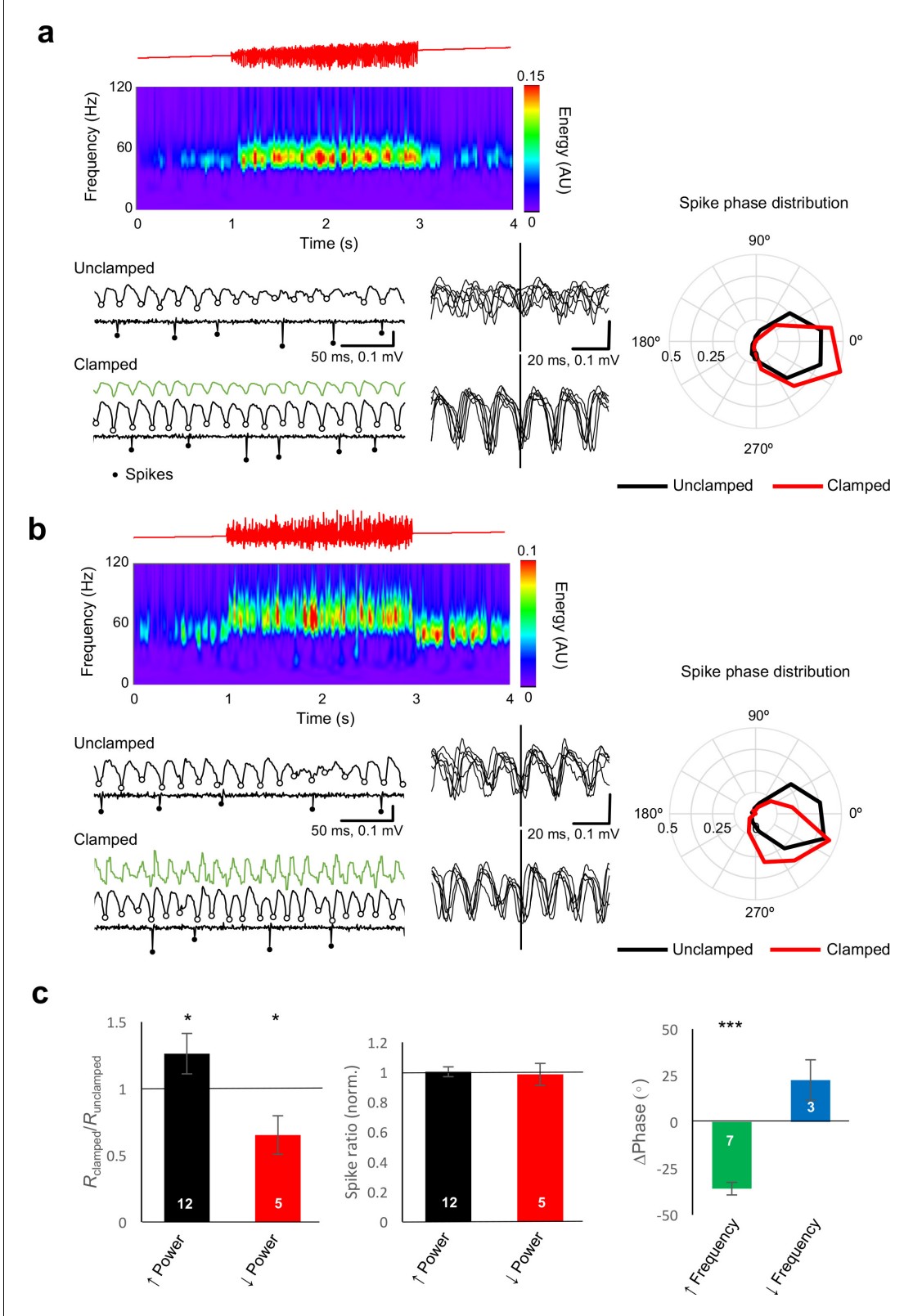

**Figure 4.** Gamma clamp alters the synchrony and phase, rather than rate, of principal cell firing . (a) Example closed loop modulation increasing gamma power. Top: red trace showing ramp command. Middle: spectrogram. Bottom: sample traces before (Unclamped) and during (Clamped) closed-loop modulation, showing the LFP and the cell-attached recording with identified spikes highlighted. LFP troughs are indicated by open circles. Six representative LFP traces, aligned by spike time, are shown at right. The polar plot indicates the distribution of spike phase for unclamped (black)
*Figure 4 continued on next page*

*Figure 4 continued*

and clamped (red) periods (averaged from 32 trials). The circular histograms sample spikes in 30° bins, and show a decrease in dispersion of spike phase during gamma clamp (LFP trough = 0°). (**b**) Example closed loop modulation increasing gamma frequency, plotted as for (a). The polar plot indicates phase advance of spiking. (**c**) Left: Bidirectional changes in power were associated with corresponding changes in the vector length (R) obtained by averaging all spike phases. This is consistent with a decrease in phase dispersion observed with an increase in power, and conversely, an increase in phase scatter with a decrease in power. Changes in power, however, did not affect the average rate of spiking, when compared with trials when gamma clamp was not applied (middle). Right: increased gamma frequency was associated with a significant phase advance of spiking. *p<0.05; ***p<0.001. Numbers of experiments are indicated in the bars.

DOI: https://doi.org/10.7554/eLife.38346.008
The following source data is available for figure 4:

**Source data 1.** *Figure 4* source data
DOI: https://doi.org/10.7554/eLife.38346.009

the unclamped situation (*Figure 4a*). Conversely, a decrease in power was accompanied by a relative desynchronization of pyramidal cell firing. This relationship was qualitatively consistent, as indicated by the change in vector length obtained from the circular average of spike phases (*Figure 4c*). The vector length increased when oscillation power increased (p=0.02, n = 12, sign test), and decreased when oscillation power decreased (p=0.025, n = 5). Strikingly, however, there was no change in the overall firing rate of pyramidal cells when the oscillation power was increased or decreased by the clamp. Changes in power were thus achieved by tightening the synchrony of firing, or by desynchronizing action potentials, rather than by altering the overall activity of pyramidal neurons.

Increases in oscillatory frequency were accompanied by a phase advance of pyramidal cell firing relative to the LFP (*Figure 4b,c*, p=$8 \times 10^{-7}$, n = 7, Hotellier test – *Zar, 2009*). A trend for a phase delay was observed in a small number of experiments where frequency-lowering clamp was tested (n = 3). This observation is consistent with the view that changes in the phase of pyramidal neuron action potentials are causally upstream of changes in gamma frequency, even though the current generators of the LFP itself are dominated by GABAergic signaling (*Gulyás et al., 2010*; *Hájos et al., 2004*; *Oren et al., 2010*).

## Excitatory current phase in principal cells determines changes in gamma spectral properties

In the examples illustrated in *Figure 4a and b*, the optogenetic modulation was applied with a phase advance over the LFP of ~0° and ~45° respectively. Why does in-phase modulation result in an increase in power, and phase-advanced modulation result in an increase in frequency? To gain a mechanistic insight into how gamma clamp operates, we examined the phase of excitation experienced by pyramidal neurons during different clamp regimes.

We repeated experiments as above, but with one pipette used to voltage–clamp a pyramidal neuron at the estimated GABA$_A$ reversal potential (approximately –70 mV), and the other pipette to record the LFP. We then measured the inward current at each phase of the gamma oscillation, as defined by the LFP, and repeated this over consecutive cycles to obtain an average time-course (*Figure 5a*). The minimum (that is, least negative) inward current during the average cycle was subtracted to yield an estimate of the phasic excitatory current, which could then be represented as a vector representing its average phase and amplitude (*Figure 5b*). During unclamped gamma, the excitatory current was small, and its average phase relative to the LFP varied among experiments, as expected from the very sparse synaptic connectivity among pyramidal neurons in CA1 (*Deuchars and Thomson, 1996*). Gamma clamp imposed a large phasic inward current (*Figure 5b*). Subtracting the vector representing the baseline phasic inward current yielded a vector representing the net excitatory current imposed by the gamma clamp (ΔE). This lagged behind the LED modulation, reflecting in part the opsin activation and deactivation kinetics (*Figure 5c*, and arrows in *Figure 5b*, right). For the example illustrated in *Figure 5* an 83° phase advance of the LED over the LFP resulted in ΔE with mean phase of 247°, where the LFP trough is defined as 0°. This yielded an increase in frequency and decrease in power of the gamma oscillation.

Comparing across different clamp regimes reveals how gamma frequency and power change in relation to the phasic excitation experienced by principal cells (*Figure 6a,b*). An increase in gamma frequency was achieved when the average excitatory current phase occurred during the down-stroke

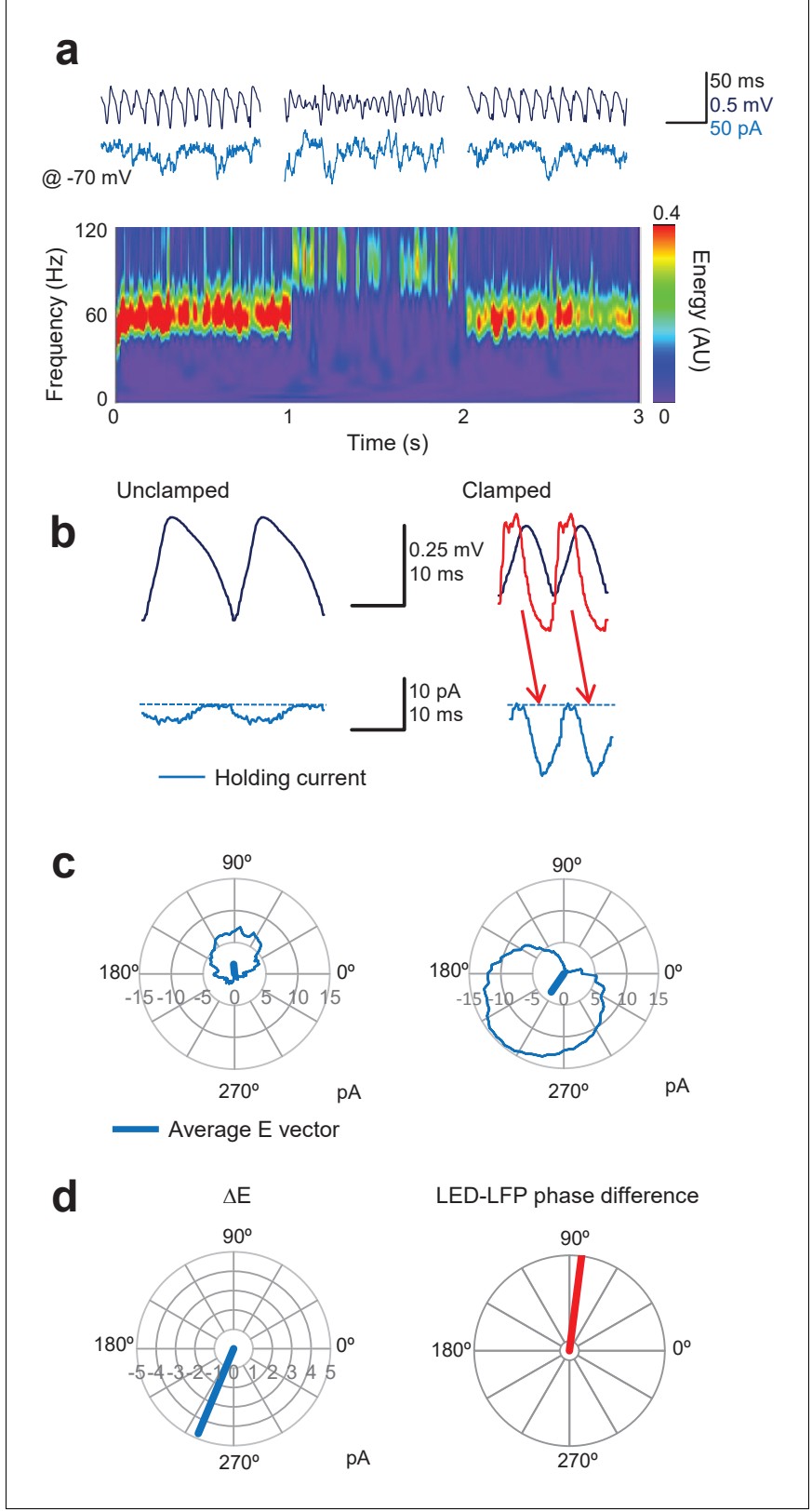

**Figure 5.** Gamma clamp imposes a phasic excitatory current to pyramidal neurons. (**a**) Top: sample LFP (navy) and simultaneously recorded holding current in one pyramidal neuron held at –70 mV (blue) before, during and after feedback modulation increasing oscillatory frequency. Bottom: spectrogram. (**b**) Two cycles of the average LFP

*Figure 5 continued on next page*

*Figure 5 continued*

waveform and membrane current without (Unclamped) and with gamma clamp (Clamped). The average phase-advanced LED command during feedback modulation is shown superimposed (red). The minimum (least negative) inward current was subtracted (dashed lines) to estimate the phasic excitation. The red arrows indicate the temporal relationship between the peak LED driver command and the maximal excitatory current. Bottom: polar plots indicating the cycle-average of the excitatory current during unclamped (left) and clamped (right) periods of the trial shown in (a). The vectors indicate the average phases of the currents. (c) Left: difference vector obtained from the vectors in (b), representing the net phasic excitatory current imposed by gamma clamp. Right: phase difference between LED and LFP for the same experiment.

DOI: https://doi.org/10.7554/eLife.38346.010
The following source data is available for figure 5:

**Source data 1.** *Figure 5* source data
DOI: https://doi.org/10.7554/eLife.38346.011

of the LFP (~180° to 360°), whilst a decrease in frequency was achieved when excitation was applied during the upstroke (~0° to 180°). An increase in power, on the other hand, was achieved with excitation around the trough (~270° to 90°), and a decrease in power occurred with excitation around the peak (~90° to 270°). Given that, under baseline conditions, pyramidal neurons fire maximally near to the trough of the LFP (0°), these data imply that the increase in frequency occurs because they are brought to firing threshold earlier (see also *Figure 4b,c*). An increase in power, on the other hand, occurs because pyramidal neurons are synchronized by adding a depolarization when they are most likely to fire (see also *Figure 4a,c*).

## Gamma clamp affects inhibitory currents

Finally, we asked how inhibitory currents in a subset of pyramidal neurons are altered by the closed-loop optogenetic manipulation, by voltage-clamping them around the glutamate reversal potential (0 mV). The mean phase and amplitude of outward inhibitory currents were calculated in a similar way, by subtracting the minimal current from the circular average of the outward GABA$_A$-receptor-mediated current (*Figure 7*). In contrast to excitatory currents, phasic inhibitory currents under baseline conditions were large, consistent with the major role of feedback interneurons in gamma (*Figure 7a,b*). Changes in inhibitory currents ($\Delta$I) were relatively smaller than for excitatory currents and were dominated by effects on the power of the oscillation. Thus, for a +83° LED-LFP phase advance, which led to a decrease in gamma power and increase in frequency (same cell as in *Figure 5*), there was little change in the average phase of the inhibitory current, although it was decreased in amplitude (*Figure 7c*). Aligning the average currents by the LFP across different trials whilst the cell was held at −70 mV or 0 mV, and during unclamped and clamped periods, yielded an insight into the relationship between excitatory and inhibitory conductances during the oscillatory cycle (*Figure 7d*). For the example shown in *Figure 5* and *Figure 7a–c*, the effective cycle changed from one dominated by phasic inhibition to one with similar relative amplitudes of phasic inhibition and excitation, with excitation leading inhibition in both cases (*Figure 7d*, left). In contrast, for a 133° LED-LFP phase delay, which led to a decrease in frequency and increase in power, the inhibitory current increased (*Figure 7d*, **right**). Similar results were obtained in eight cells.

## Discussion

The present study shows that closed-loop optogenetic manipulation of principal cells allows predictable, bidirectional and dissociable changes in the power and frequency of gamma oscillations. We observed a broad consistency between the frequency manipulation achieved with closed-loop optogenetic feedback and that predicted from the phase response behavior previously observed with intermittent optogenetic stimuli (*Akam et al., 2012*). Optogenetically and pharmacologically induced gamma also exhibited similar dynamical properties in that study, implying that the principles uncovered in the present work are not specific to the way gamma oscillations were elicited.

Previous studies have stressed the importance of fast-spiking parvalbumin-positive (PV+) interneurons in gamma (*Cardin et al., 2009*; *Sohal et al., 2009*) (but see (*Veit et al., 2017*)). PV+ basket cells tend to fire with very little phase dispersion, close to one-to-one with each cycle of the

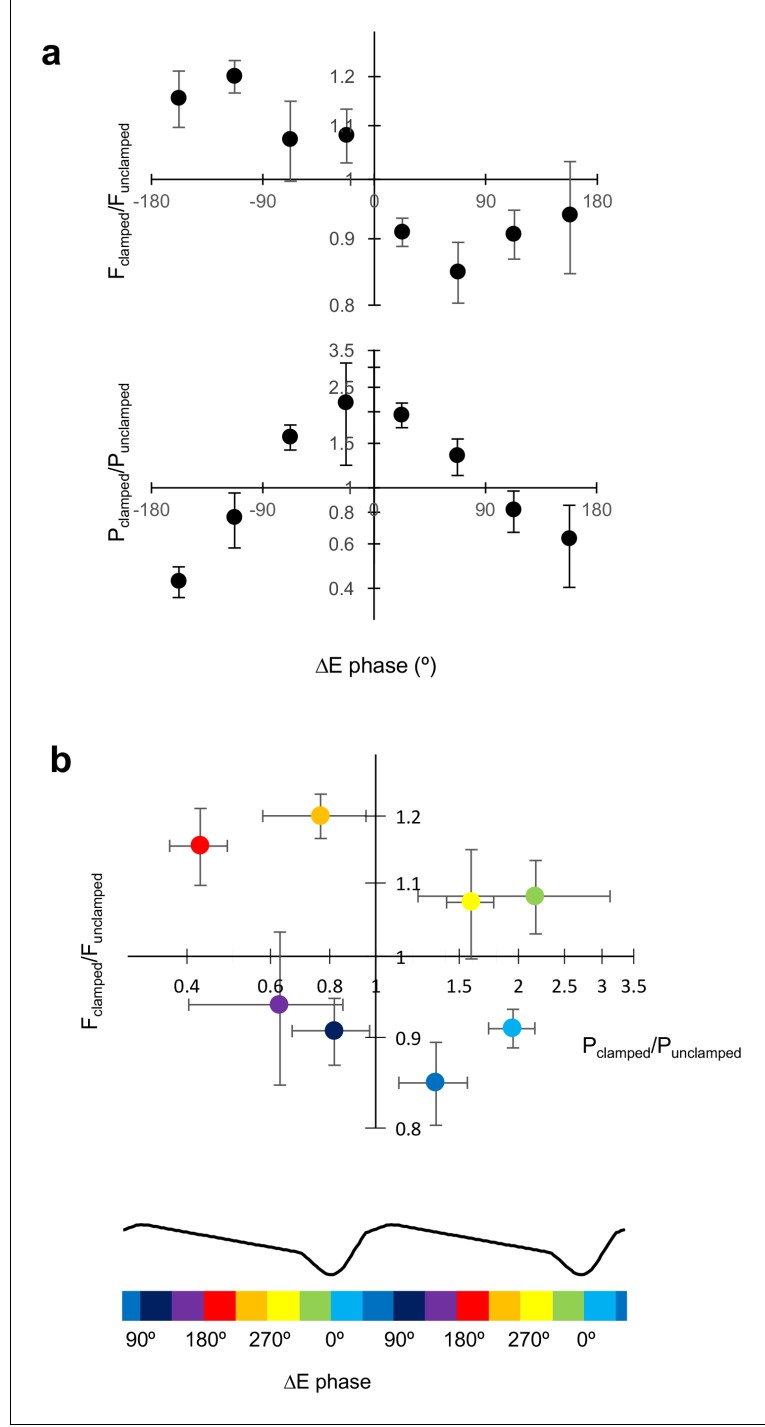

**Figure 6.** Excitatory current phase determines changes in frequency and power. (**a**) Change in gamma frequency and power, plotted against the phase of the net excitatory current (ΔE) calculated as in *Figure 5*. Confidence intervals are SEM (n = 13). (**b**) $F_{clamped}/F_{unclamped}$ plotted against $P_{clamped}/P_{unclamped}$ for different excitatory current phases, indicated by the colour code below, aligned with the average LFP waveform. Pyramidal neurons spike around the trough of the LFP (0°).

DOI: https://doi.org/10.7554/eLife.38346.012

The following source data is available for figure 6:

**Source data 1.** *Figure 6* source data
DOI: https://doi.org/10.7554/eLife.38346.013

oscillation in vitro (*Bartos et al., 2007*; *Gulyás et al., 2010*). Our attempts to achieve gamma clamp by targeting interneurons rather than pyramidal cells have thus far been unsuccessful because their out of phase recruitment powerfully suppresses the oscillation (data not shown). The weaker phase-locking of pyramidal than PV+ cell firing to gamma oscillations, together with their sparse firing on successive cycles of gamma (*Csicsvari et al., 2003*; *Gulyás et al., 2010*; *Tukker et al., 2007*), may however confer a broader dynamic range over which they can influence the phase, frequency and amplitude of the oscillation. Taken together with previous evidence that open-loop sinusoidal opto-genetic stimulation of principal cells can entrain a gamma oscillation (*Akam et al., 2012*), the present data underline the importance of action potential timing in principal cells in the spectral and temporal properties of hippocampal gamma, notwithstanding the evidence that the LFP itself is dominated by inhibitory currents in principal cells (*Oren et al., 2010*), and argue against a model where the function of principal cells is only to depolarize a population of reciprocally connected interneurons.

Closed-loop manipulations have been applied previously in the context of network oscillations, using either electrical and optogenetic stimuli delivered at specific phases of theta or gamma oscillations, in order to probe the mechanisms of long-term plasticity induction (*Huerta and Lisman, 1995*; *Pavlides et al., 1988*) or sharp-wave ripple generation (*Stark et al., 2014*), or to test the theta phase-dependence of memory encoding and retrieval (*Siegle and Wilson, 2014*). A similar strategy has been used to interrupt experimental thalamocortical seizures (*Berényi et al., 2012*). However, these studies have not aimed at modulating the amplitude or frequency of an on-going oscillation.

We have focused on gamma because a local circuit is sufficient to generate the oscillation, and we have previously shown that the phase response behaviour of hippocampal gamma is well described by a simple dynamical model (*Akam et al., 2012*). The circuits underlying theta and other oscillations either involve longer-range connections in the brain or are poorly defined. They are therefore less likely to be amenable to local optogenetic manipulation. This does not exclude the possibility that, for instance, theta oscillations in the hippocampus could be manipulated by closed-loop modulation of excitability in the basal forebrain.

The ability to alter the amplitude and frequency of gamma suggests a versatile tool to test the roles of gamma in information routing and other high-level brain functions, both in health and in disease states such as schizophrenia (*Uhlhaas and Singer, 2010*). Hitherto, most experimental manipulations of oscillations have relied on periodic stimulation, which can entrain network oscillations (*Akam et al., 2012*) or evoke oscillations in an otherwise asynchronous network (*Cardin et al., 2009*; *Sohal et al., 2009*). Transcranial stimulation designed to entrain oscillations in vivo can bias perception (*Neuling et al., 2012*; *Romei et al., 2010*; *Thut et al., 2011*) and bidirectionally affect performance in motor (*Joundi et al., 2012*) and working memory (*Polanía et al., 2012*) tasks. However, external periodic stimulation is not well suited to desynchronize network activity or to suppress oscillatory dynamics. Furthermore, if periodic stimulation is used, the desired change in amplitude or frequency is achieved at the cost of imposing an externally determined phase on the oscillation. This will prevent the oscillation from entraining to endogenous periodic signals such as those arising from other oscillating networks or periodic sensory stimuli.

Closed-loop stimulation, in which signals recorded from a network are used in real time to bias its state, in principle provides an alternative way of manipulating network oscillations, and has been used to interfere with pathological rhythms in models of Parkinson's disease (*Rosin et al., 2011*), to suppress Parkinsonian tremor (*Brittain et al., 2013*), and in a model of thalamocortical epilepsy (*Butt et al., 2005*). This approach relies on an artificial feedback loop which either counteracts or amplifies the endogenous feedback responsible for synchronizing the network (*Rosenblum and Pikovsky, 2004*). Importantly, optogenetics has the advantage over electrical stimulation that the modulation can be distributed across a population of neurons. We have, moreover, shown that closed-loop manipulation of a gamma oscillation can be achieved without a net increase or decrease in the average firing rate of neurons, implying that it would not necessarily perturb information represented as an average firing rate code.

Extrapolating from in vitro gamma to the brain in situ presents several technical challenges, including the need for optical fibers to illuminate the tissue and the potential for photoelectrical artifacts. Moreover, oscillations are generally less prominent because the current generators from multiple oscillating and non-oscillating populations overlap, complicating the evaluation of phase and frequency. Nevertheless, the present study identifies some general principles to guide attempts to

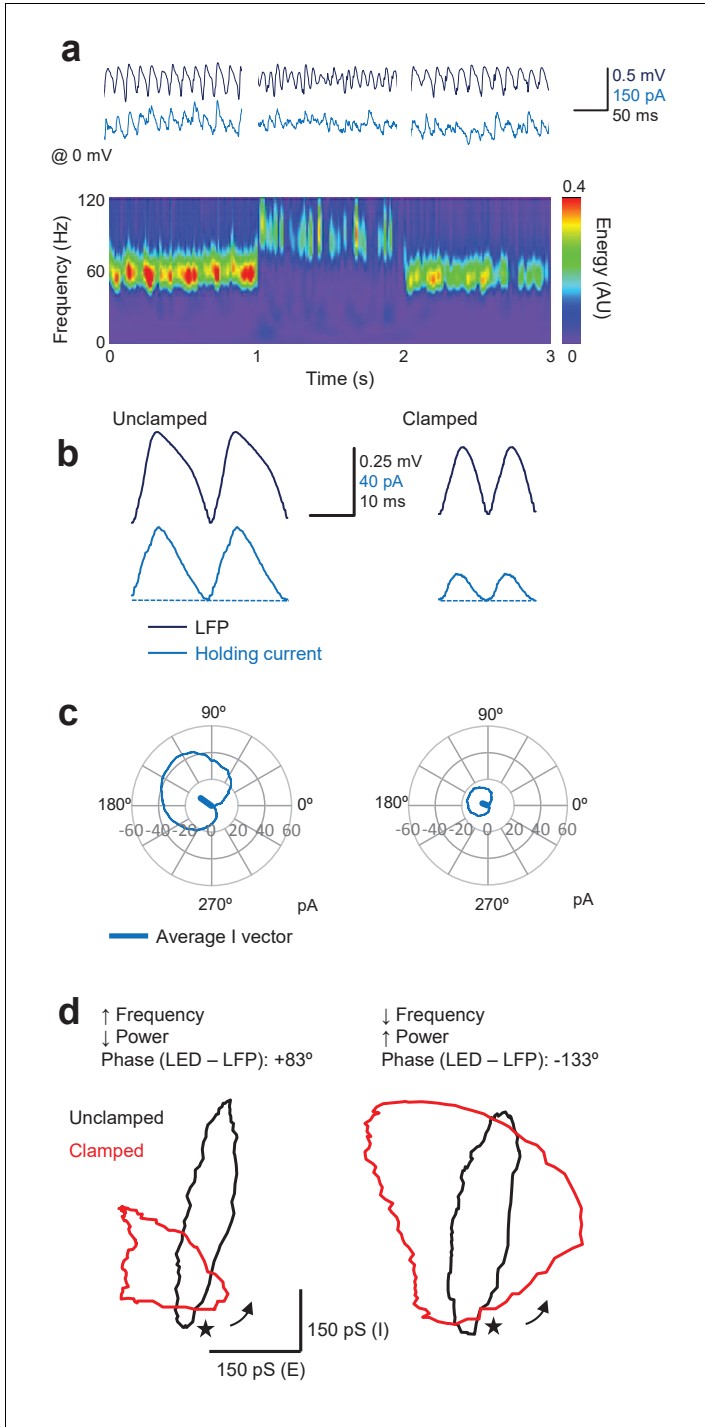

**Figure 7.** Gamma clamp afffects phasic inhibitory currents in pyramidal neurons. (**a**) Top: sample LFP (navy) and simultaneously recorded holding current in one pyramidal neuron held at 0 mV (blue) before, during and after feedback modulation increasing oscillatory frequency. Bottom: spectrogram. Same cell as in *Figure 5*. (**b**) Two cycles of the average LFP waveform and membrane current without (Unclamped) and with gamma clamp (Clamped). The average phase-advanced LED command during feedback modulation is shown superimposed (red). The minimum (least positive) outward current was subtracted (dashed lines) to estimate the phasic inhibition. (**c**) Polar plots indicating the cycle-average of the inhibitory current during unclamped (left) and clamped (right) periods of the trial shown in (**a**). The vectors indicate the average phases of the currents. (**d**) Left: phasic inhibition (vertical scale) plotted against phasic excitation (horizontal scale) estimated by aligning cycle-average membrane conductances by the LFP without (black) and with (red) oscillation clamp with +83° LED-LFP phase advance. Same

*Figure 7 continued on next page*

*Figure 7 continued*

cell and clamp regime as shown in (**a–c**). Right: representative example of excitatory and inhibitory conductances obtained with 133° LED-LFP phase delay. Same cell but different clamp regime. The cycles are arbitrarily anchored to the trough of the LFP (*). The arrows indicate the direction of the excursion.

DOI: https://doi.org/10.7554/eLife.38346.014

The following source data is available for figure 7:

**Source data 1.** *Figure 7* source data

DOI: https://doi.org/10.7554/eLife.38346.015

achieve bidirectional and dissociable modulation of oscillatory frequency and power in vivo. This should allow a definitive test of the causal role of gamma in functions such as attention modulation and information routing (*Sohal, 2016*).

# Materials and methods

## Key resources table

| Reagent type (species) or resource | Designation | Source or reference | Identifiers | Additional information |
|---|---|---|---|---|
| Strain, strain background (*Mus musculus*) | Wild type, C57bl6 mice | UCL Biological Services | | |
| Genetic reagent | Viral vector (C1V1): AAV5-CaMKIIa-C1V1(E122T/E162T)-TS-eYFP | UNC Vector Core | Serotype: 5 C1V1: AAV-CaMKIIa-C1V1(E122T/E162T)-TS-EYFP | |
| Other | Equipment (viral injections): Sterotaxic frame | Kopf Instruments | Model 900 | |
| Chemical compound, drug | Salts, drugs | Sigma | | |
| Other | Equipment (LED light source) | Cairn Instruments | OptoLED, 590 nm | |
| Other | Equipment (LED light source) Custom assembled LED | Thorlabs | High-power mounted LED: M590L2; Tube lens: SM1V10; Planoconvex lens: LA1951-A-N-BK7; Coupler: SM1T2; Adapter for microscope: SM1A14 | |
| Other | Equipment (LED driver) | Thorlabs | DC2100 | |
| Other | Equipment (Upright microscope) | Olympus | BX51WI, UMPLFLN 20X W IR | |
| Other | Equipment (Upright microscope) | Scientifica | SliceScope, UMPLFLN 20X W IR | |
| Other | Equipment (Amplifier) | Molecular Devices | Multiclamp 700B | |
| Other | Equipment (Data acquisition card) | National Instruments | PCI-6221 | |
| Other | Equipment (computing) Real-time controller | National Instruments | cRIO-9022 | |

*Continued on next page*

*Continued*

| Reagent type (species) or resource | Designation | Source or reference | Identifiers | Additional information |
|---|---|---|---|---|
| Other | Equipment (computing). FPGA | Xilinx | Virtek-5 | |
| Software, algorithm | LabVIEW | National Instruments | *LabVIEW, LabVIEW Real-Time* 2013, 2014, 2015, 2016, 2017 | Custom virtual instruments |
| Software, algorithm | *R* | www.R-project.org | Ver. 3.3.0 for Mac | |

All procedures followed the Animals (Scientific Procedures) Act, 1986, and were reviewed by the UCL Institute of Neurology Animal Welfare and Ethical Review Body. P21 male C57 mice were anesthetized with isoflurane and placed in a stereotaxic frame (Kopf Instruments). A suspension of AAV5-CaMKIIa-C1V1(E122T/E162T)-TS-eYFP (UNC Vector Core, titre $5 \times 10^{12}$ IU/ml) was injected at a rate of 100 nl/min into four sites in both hippocampi (injection volume: 300–500 nl per site). The antero-posterior injection coordinate was taken as 2/3 of the distance from bregma to lambda. The lateral coordinates were 3.0 mm from the midline, and the ventral coordinates were 3.5, 3.0, 2.5 and 2.0 mm from the surface of the skull.

Hippocampal slices were prepared at least 4 weeks later. Animals were sacrificed by pentobarbitone overdose and underwent transcardiac perfusion with an oxygenated solution containing (in mM): 92 N-methyl-D-glucamine-Cl, 2.5 KCl, 1.25 $NaH_2PO_4$, 20 HEPES, 30 $NaHCO_3$, 25 glucose, 10 $MgCl_2$, 0.5 $CaCl_2$, 2 thiourea, 5 Na-ascorbate and 3 Na-pyruvate, with sucrose added to achieve an osmolality of 315 mOsm/L. Brain slices (400 µm thick) were prepared at room temperature and then incubated at 37°C for 12 min in the same solution. They were subsequently stored at room temperature, in a solution containing (in mM): 126 NaCl, 3 KCl, 1.25 $NaH_2PO_4$, 2 $MgSO_4$, 2 $CaCl_2$, 24 $NaHCO_3$, 10 glucose, shielded from light, before being transferred to the stage of an upright microscope (Olympus BX51WI or Scientifica SliceScope), where they were perfused on both sides with the same solution at 32° C. Expression of C1V1 in CA1 was verified by epifluorescence, and CA3 was ablated to focus on local gamma-generating mechanisms.

Epifluorescence imaging and C1V1 stimulation were achieved with LEDs (OptoLED, Cairn Instruments, or assembled from Thorlabs components using an M590L2 590 nm LED and a DC2100 high-power LED driver). The light source was coupled to the epifluorescence illuminator of the microscope, with a silver mirror in the place of a dichroic cube. Wide-field illumination was delivered via a 20x, 0.5 NA water immersion objective. The current delivered to the LED was kept in the linear input-output range, and the irradiance was <5 mW/mm². Light ramps typically lasting 8 s were delivered every 30 – 45 s.

LFPs were recorded in the CA1 pyramidal layer using patch pipettes filled with extracellular solution and a Multiclamp 700B amplifier (Molecular Devices), and band-pass filtered between 1 and 200 or 500 Hz. A linear LED ramp command was generated via a multifunction data acquisition card (National Instruments PCI-6221) and, together with the LFP, was digitized using a real-time controller (National Instruments cRIO-9022) with a Xilinx Virtex-5 FPGA (cRIO-9133) operating at a loop rate of 10 kHz. The ramp was multiplied by ($1 + k_1$LFP $+ k_2 d$LFP/$d$t), stepping through different values of $k$ in a pseudo-random order for successive trials. $d$LFP/$d$t was calculated as the difference between successive digitization values in the FPGA, averaged over successive 2 ms intervals to minimize high-frequency noise. The output of the FPGA/real-time controller was sent to the LED driver, and digitized in parallel with the LFP at 10 kHz on the data acquisition PC.

To study the phase relationship of action potentials and the LFP oscillation, a cell-attached recording was obtained using a second patch pipette held in voltage clamp mode, low-pass filtered at 10 kHz and digitized in parallel with the LFP and LED command signal. The phasic excitatory or inhibitory current was recorded in the same way, but using a whole-cell pipette containing (in mM): K-gluconate (145), NaCl (8), KOH-HEPES (10), EGTA (0.2), Mg-ATP (2) and Na 3 -GTP (0.3); pH 7.2; 290 mOsm. Phasic conductances (*Figure 7d*) were estimated from Ohm's law, assuming a driving force of 70 mV.

Off-line analysis was performed in LabVIEW (National Instruments) and R. Time-frequency spectrograms were calculated using a Morlet wavelet transform and are displayed as heat maps. Because the gamma oscillation was non-stationary, its frequency was estimated by calculating the short-term Fourier transform and then averaging the mean instantaneous frequency for successive overlapping intervals. The power of the oscillation was estimated in the same way, by averaging the power at the mean instantaneous frequency.

Spikes were identified using threshold crossing. The instantaneous oscillation phase was estimated by passing a 200 ms segment of the LFP centered on the spike through a Hanning window, and then calculating its phase and frequency using the Extract Single Tone VI in LabVIEW.

To estimate the phase relationship between spikes or membrane currents and the gamma oscillation, we first identified successive troughs of the LFP using the WA Multiscale Peak Detection VI in LabVIEW. Gamma cycles that deviated more than 20% from the modal period were rejected. The membrane current waveform between successive troughs was then expressed as a function of instantaneous phase and averaged over all accepted cycles in the interval. The minimal (least negative) inward current recorded at −70 mV was subtracted to yield the average phasic excitatory current waveform. To estimate the phasic inhibitory current waveform, the minimal outward current was subtracted whilst holding cells at 0 mV.

## Acknowledgements

We are grateful to Francis Carpenter, Flora Lee and Iris Oren for pilot experiments, and to Karl Deisseroth for sharing C1V1. This work was supported by the Wellcome Trust.

## Additional information

### Funding

| Funder | Grant reference number | Author |
|--------|------------------------|--------|
| Wellcome | 095580/Z/11/Z | Dimitri Michael Kullmann |
| Wellcome | WT104033AIA | Dimitri Michael Kullmann |

The funders had no role in study design, data collection and interpretation, or the decision to submit the work for publication.

### Author contributions

Elizabeth Nicholson, Dmitry A Kuzmin, Formal analysis, Investigation, Writing—review and editing; Marco Leite, Formal analysis, Writing—review and editing; Thomas E Akam, Conceptualization, Writing—review and editing; Dimitri Michael Kullmann, Conceptualization, Software, Formal analysis, Supervision, Funding acquisition, Investigation, Methodology, Writing—original draft, Project administration, Writing—review and editing

### Author ORCIDs

Thomas E Akam (iD) https://orcid.org/0000-0002-1810-0494
Dimitri Michael Kullmann (iD) http://orcid.org/0000-0001-6696-3545

### Ethics

Animal experimentation: This study was performed in accordance with the Animals (Scientific Procedures) Act, 1986, and were reviewed by the UCL Institute of Neurology Animal Welfare and Ethical Review Body.

### Decision letter and Author response

Decision letter https://doi.org/10.7554/eLife.38346.018
Author response https://doi.org/10.7554/eLife.38346.019

## Additional files

### Supplementary files
• Transparent reporting form
DOI: https://doi.org/10.7554/eLife.38346.016

### Data availability
Source data files have been provided for Figures 1-7.

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
