## [Decision Letter]

Thank you for submitting your article "Analog closed-loop optogenetic modulation of hippocampal pyramidal cells dissociates γ frequency and amplitude" for consideration by *eLife*. Your article has been reviewed by three peer reviewers, one of whom is a member of our Board of Reviewing Editors, and the evaluation has been overseen by Michael Frank as the Senior Editor. The following individual involved in review of your submission has agreed to reveal his identity: Imre Vida (Reviewer #2). Two other reviewers remain anonymous.

The reviewers have discussed the reviews with one another and the Reviewing Editor has drafted this decision to help you prepare a revised submission.

Summary:

Nicholson et al. used an optogenetic approach to examine the influence of phasic excitation of pyramidal cells (PCs) on the spectral and temporal dynamics of γ oscillations in acute slices of the hippocampal CA1 area. The authors demonstrate that phase-advancing or phase-delaying excitatory inputs onto PCs alter the synchrony but not the frequency of action potentials generated in CA- PCs. The authors recorded action potentials from pyramidal cells in the cell attached mode to examine how γ-clamp affects the timing and rate of action potentials generated by pyramidal cells. A decrease in oscillatory power was accompanied by a desynchronization of pyramidal cell activity whereas an increase in power was associated with an increase in the synchrony of PC discharges. The overall activity did not contribute to network synchrony. Increase in oscillatory frequency was associated with a phase advance of PC firing but in-phase modulation resulted in an increase in oscillatory power. Next, the authors examined the potential mechanisms underlying the changes in frequency and power of γ oscillations by recording excitatory currents from an individual PC during network oscillations. They observed that an increase in PC discharges was achieved when PCs received excitatory inputs earlier (~200°) than the expected PC firing. Thereby, PCs were brought above threshold earlier during the γ cycle. Finally, the authors demonstrated that oscillatory frequency could be reduced if the excitatory input arrived at the peak of PC activity. Although the manuscript is very well written and the study well performed, the reviewers formulated some concerns which the authors should address.

Essential revisions:

1) Although it was examined how excitatory currents are changed as a result of the closed loop situation, it is unclear how inhibitory currents changed. The authors suggest that increased excitation may enable pyramidal cells to fire earlier relative to the periodic inhibition. It is possible, however, that the phase of inhibition also changed relative to the LFP considering the increased contribution of the excitatory currents to the LFP. To fully understand how the γ circuit oscillator functioned during the frequency increase would require describing the timing of inhibition as well relative to the LFP.

2) Perhaps the authors could provide an explanation of how the frequency of oscillations could be reduced by their manipulations. Perhaps, by extending the firing of pyramidal cells, interneuron activity could be also prolonged and delayed? Again understating inhibitory dynamics would give a more complete picture about the state of the circuit in generating different frequency γ-band oscillations.

3) Is the difference in the shape of the LED command depending on the contribution of the original LFP signal (k1) and its first derivative (k2) (e.g. Figure 2 red traces). This feature of the feedback signal makes systematic analysis or comparison of phase dependent changes quantitatively difficult, if not impossible.

4) Finally, another related aspect which needs clarification is the impact of the relatively slow kinetics of the optogenetic actuator. As illustrated in Figure 5 this could result in a phase difference/delay of over 180 degrees. Arguably, this phase difference is frequency dependent. In view of this property of the feedback optogenetic excitation, how does the analysis of LED-LFP phase relationship presented in Figure 2 and 3 help the reader to understand the impact and the underlying mechanisms? Clearly, the authors are aware of this issue and present Figure 6 which supersedes Figure 3 and puts the initial conclusions into a new perspective. This aspect needs to be clear to the reader from the beginning on.

---

## [Author Response]

Essential revisions:1) Although it was examined how excitatory currents are changed as a result of the closed loop situation, it is unclear how inhibitory currents changed. The authors suggest that increased excitation may enable pyramidal cells to fire earlier relative to the periodic inhibition. It is possible, however, that the phase of inhibition also changed relative to the LFP considering the increased contribution of the excitatory currents to the LFP. To fully understand how the γ circuit oscillator functioned during the frequency increase would require describing the timing of inhibition as well relative to the LFP.

We recorded inhibitory currents in a subset of cells by holding them at the glutamate reversal potential (~0 mV). Under baseline conditions (that is, in the absence of closed-loop oscillation clamp), the phasic inhibitory currents were much larger than the phasic excitatory currents, as expected from the major role of recurrent inhibition in γ and low recurrent excitatory synaptic connectivity in CA1. The fractional change in amplitude of phasic inhibitory currents with γ clamp was, on average, less than 20% of the fractional change in amplitude of excitatory currents. The change in phase was also much smaller than the change in phase of the excitatory currents. This made it difficult to apply the same analysis procedure as for the excitatory currents. That is, subtracting the vector representing the average unclamped inhibitory current from the vector representing the average clamped inhibitory current in some cases resulted in a vector that pointed in a phase direction that was difficult to interpret intuitively. However, a robust outcome was a tendency for changes in the amplitude of the inhibitory current to covary with changes in the power of the oscillation. A simple mechanistic interpretation is that, by modulating excitatory currents, the optogenetic drive alters the timing of firing of principal neurons, which as a downstream consequence affects the recruitment of interneurons, resulting in parallel changes in γ power and inhibitory currents. We have now added a section summarizing the analysis of inhibitory currents to the end of the Results, with representative examples in Figure 7.

2) Perhaps the authors could provide an explanation of how the frequency of oscillations could be reduced by their manipulations. Perhaps, by extending the firing of pyramidal cells, interneuron activity could be also prolonged and delayed? Again understating inhibitory dynamics would give a more complete picture about the state of the circuit in generating different frequency γ-band oscillations.

As outlined above, changes in inhibitory current amplitudes appear to track changes in oscillatory power, with relatively smaller changes in phase than the changes in excitatory current. In the original version we allowed an inconsistency in the definition of LFP phase between different sections of the paper. We have now streamlined this so that it is clear that 0° refers to the trough of the LFP, which coincides with the maximal firing probability of pyramidal neurons. The conclusion of Figure 6B is now more intuitive: applying an excitatory drive during the downstroke of the LFP (180° to 360°) phaseadvances the firing of pyramidal neurons and therefore results in an increase in frequency. Conversely, applying an excitatory drive during the upstroke (0° to 180°), with a corresponding decrease in excitation during the downstroke, retards pyramidal neuron firing and decreases the oscillatory frequency. This does not mean that interneurons play no role, but by targeting principal cells with the optogenetic drive, changes in their recruitment is mechanistically downstream.

3) Is the difference in the shape of the LED command depending on the contribution of the original LFP signal (k1) and its first derivative (k2) (e.g. Figure 2 red traces). This feature of the feedback signal makes systematic analysis or comparison of phase dependent changes quantitatively difficult, if not impossible.

We agree that because the formula results in spectral distortion, the phase difference between the LFP and the LED for some combinations of k1 and k2 was only an approximation. For want of a better method, we calculated it simply by taking the phase at the frequency corresponding to the maximal magnitude of the cross spectrum. The averaged traces in Figure 2C, D actually accentuate the apparent ambiguity in the phase calculation because they give the impression that each cycle of the LFP was highly stereotyped. In fact, the γ oscillation always fluctuates from cycle to cycle in amplitude and phase, and so any error arising from spectral distortion is unlikely to have a major impact on the conclusions. We now acknowledge that the estimated phase difference was only approximate in the text.

4) Finally, another related aspect which needs clarification is the impact of the relatively slow kinetics of the optogenetic actuator. As illustrated in Figure 5 this could result in a phase difference/delay of over 180 degrees. Arguably, this phase difference is frequency dependent. In view of this property of the feedback optogenetic excitation, how does the analysis of LED-LFP phase relationship presented in Figure 2 and 3 help the reader to understand the impact and the underlying mechanisms? Clearly, the authors are aware of this issue and present Figure 6 which supersedes Figure 3 and puts the initial conclusions into a new perspective. This aspect needs to be clear to the reader from the beginning on.

We hesitated about the order of the narrative. We opted for showing how the behaviour of the oscillation clamp accords with the phase-response curve previously reported in Akam et al., 2012, and then included the spiking data before finishing with the membrane currents, where the underlying mechanisms are resolved. We agree that this seems a long journey, and have therefore added a paragraph to explain the sequence (at the end of the section reconciling clamp behaviour with the PRC).